# Factors of Anti-Vascular Endothelial Growth Factor Therapy Withdrawal in Patients with Neovascular Age-Related Macular Degeneration: Implications for Improving Patient Adherence

**DOI:** 10.3390/jcm10143106

**Published:** 2021-07-14

**Authors:** Fumi Gomi, Reiko Toyoda, Annabelle Hein Yoon, Kota Imai

**Affiliations:** 1Department of Ophthalmology, Hyogo College of Medicine, Hyogo 663-8501, Japan; fgomi@hyo-med.ac.jp; 2Ophthalmology Medical Franchise Department, Medical Division, Novartis Pharma K.K., Tokyo 105-6333, Japan; reiko-toyoda.ac@cmic.co.jp (R.T.); annabelle.h.yoon@gmail.com (A.H.Y.)

**Keywords:** anti-VEGF therapy, age-related macular degeneration, withdrawal, adherence, patient-reported outcomes

## Abstract

We investigated the factors associated with the discontinuation of anti-vascular endothelial growth factor (VEGF) therapies in patients with neovascular age-related macular degeneration (AMD). Japanese patients with AMD aged ≥50 years, reporting at least one prior injection of an anti-VEGF drug, completed an online survey covering reasons for discontinuation or dissatisfaction with therapy, quality of life (EQ-5D-5L) and patient activation (PAM-13). The respondents were divided into two cohorts: Cohort 1—patients who discontinued anti-VEGF therapy (*n* = 207); Cohort 2—patients continuing anti-VEGF therapy (*n* = 65). The most common reason for discontinuing therapy was the “doctor’s decision” in 89.4% (Cohort 1-1). In the other 22 (10.6%) patients in Cohort 1 (Cohort 1-2), reasons included “no deterioration in vision”, “financial burden” and “ineffective treatment”. Patients in Cohort 2 were dissatisfied with “long waiting times” (77%), “financial burden” and “ineffective treatment”. Pain/discomfort posed the greatest impact on quality of life. Only 5% of patients in Cohorts 1-1 and 2 and none in Cohort 1-2 were considered advocates for their own health. In conclusion, most patients who discontinued anti-VEGF therapy did so at their doctor’s decision. Addressing the reasons associated with discontinuation or dissatisfaction with anti-VEGF therapies might help improve their continuation.

## 1. Introduction

Age-related macular degeneration (AMD) is a progressive eye disease that affects the central retina and can cause vision loss [1]. Approximately 90% of cases of severe vision loss are due to neovascular AMD (nAMD) [2]. Because choroidal neovascularization, the key contributor to vision loss in nAMD, is driven by vascular endothelial growth factor (VEGF), several anti-VEGF therapies have been developed are now recommended as first-line therapy for nAMD [3]. Because there is no definitive cure for nAMD, early detection and regular, continued follow-up are essential [4]. However, some patients discontinue anti-VEGF therapies [5,6], increasing the risk of further deterioration in visual acuity [7]. Some concerns associated with anti-VEGF therapies include the cost of treatment, fear of injection into the eye and discomfort [8,9]; however, few studies have examined why patients stop these therapies. Therefore, in this study, we performed an online survey of patients with nAMD to elucidate the major reasons for discontinuing anti-VEGF therapy. The respondents were encouraged to answer frankly in the anonymous survey. The survey also covered questions aimed at understanding the reasons why patients who continued anti-VEGF therapy were dissatisfied with their therapy, the characteristics and common comorbidities of nAMD patients, quality of life and patient activation.

## 2. Patients and Methods

### 2.1. Ethics

This survey adhered to the Japanese Ethical Guidelines for Medical and Health Research Involving Human Subjects, which encompass the ethical principles defined in the Declaration of Helsinki, as well as regulatory standards for non-interventional studies. The study protocol was reviewed by the central ethics committee at the Clinical Research Promotion Network Japan (CR-IRB-0104-1-1; 22 August 2019). All participants provided informed consent.

### 2.2. Setting

The online survey was conducted by Social Survey Research Information Co., Ltd. (SSRI Inc., Tokyo, Japan) using anonymized data from the Rakuten Insight AMD patient panel (Rakuten, Tokyo, Japan), one of 186 patient panels managed by Rakuten [10]. Patients with AMD who were registered on the Rakuten AMD panel were invited to participate in the online survey. The survey was open for 30 days, and data were collected from 11 September to 10 October 2019. Eligibility criteria for the survey were diagnosis of AMD, age ≥50 years and at least one prior intravitreal injection of an anti-VEGF drug. See the Appendix A for further information on study design.

### 2.3. Survey Design

The eligible patients were asked questions about their demographics, social history and medical history (Appendix A). Based on their self-reported medical history, the patients were classified into three cohorts: Cohort 1—patients who did not receive anti-VEGF therapy for ≥6 months at the doctor’s decision (Cohort 1-1) or at the patient’s own decision (Cohort 1-2); Cohort 2—patients who did receive anti-VEGF therapy within the previous 6 months. Cohort 1 comprised patients who were considered to have stopped treatment based on an interval of ≥6 months since their last dose. Cohort 2 comprised patients who were considered to be continuing treatment, regardless of their level of satisfaction. Patients were then asked to complete one of two questionnaires (main questionnaire) according to the cohort. Patients in Cohorts 1-1 were asked about their reasons for discontinuing anti-VEGF therapy, other than the doctor’s decision, and patients in Cohort 1-2 were asked about their reasons for discontinuing anti-VEGF therapy in the main questionnaire. The patients were provided a list of 14 predefined reasons with four possible responses (definitely true/mostly true/mostly false/definitely false) (Appendix A). Patients in Cohort 2 were asked about their reasons for dissatisfaction with anti-VEGF therapy and were given a list of 13 reasons, which were slightly modified from the questionnaire posed to Cohort 1 in the main questionnaire (Appendix A). The 14 possible reasons for discontinuation (Cohorts 1-1 and 1-2) and the 13 possible reasons for dissatisfaction (Cohort 2) were introduced in a random order to minimize question order effects. All of the patients then completed the Japanese versions of the EQ-5D-5L to assess health-related quality of life and the PAM-13 to measure patient activation levels (Appendix A). We defined discontinuation as stopping administration of intravitreal anti-VEGF injections for ≥6 months.

### 2.4. Statistical Analyses

All subjects who at least completed the main questionnaire were included in the data analyses. Patient demographics, social/medical history and reasons for discontinuation/dissatisfaction were analyzed descriptively and are reported as the number and percent of patients. The results of the EQ-5D-5L are presented descriptively [11]. PAM-13 was calculated using standardized methods by the vendor (Insignia Health) to yield four levels of activation [12]. All analyses were conducted by SSRI, using BellCurve for Excel and R, as appropriate. Pearson’s χ^2^ test, Fisher’s exact test, *t* test and analysis of variance were used to calculate *p* values for exploratory purposes. The four response categories of the main questionnaire were collapsed into two categories (agree/disagree) for analyses. Because of the exploratory nature of the study, no adjustment for multiple testing was performed. The results of the analysis should be used to generate hypotheses and are not to be considered confirmatory.

## 3. Results

### 3.1. Patient Stratification and Characteristics

Overall, 272 patients completed the survey (Figure 1), with 207 patients in Cohort 1 (discontinued treatment) and 65 in Cohort 2 (continuing treatment). In Cohort 1, 185 (89%) discontinued anti-VEGF therapy due to their doctor’s decision (Cohort 1-1) and 22 (11%) discontinued due to their own decision (Cohort 1-2). Half (93/185; 50%) of patients in Cohort 1-1 thought their treatment was complete; the others thought they were in a period of monitoring. The characteristics of the three cohorts are summarized in Table 1. The best-corrected visual acuity was generally similar in all three cohorts (Table 2).

### 3.2. Treatment-Related Components

Patients in Cohort 1-2 were more likely to visit an eye clinic than patients in Cohorts 1-1 and 2 (*p* = 0.0402). Public transport and car were the most common means of transport to hospital (Table 3). The majority of patients in Cohort 1 had been treated for <1 year (58% and 64% in Cohorts 1-1 and 1-2, respectively, vs. 17% in Cohort 2; *p* < 0.001, Table 3). Furthermore, 31% of patients in Cohort 1-1 received monthly injections, and most of them (41/57; 72%) continued treatment for <1 year. Approximately 30% of patients in Cohort 1 were treated for <3 months, indicating that treatment was discontinued during the loading doses. In addition, 16% (33/207) of patients in Cohort 1 reported <3 months’ experience of treatment with an unknown treatment frequency, suggesting that 16% of patients in Cohort 1 discontinued treatment after the first injection.

The costs (out-of-pocket payments) were widely distributed across the categories. Cohort 2 had fewer patients in the category of JPY ≥ 50,000 (approx. USD 450; *p* = 0.0149). Overall, 26% of patients reported out-of-pocket payments of JPY ≥ 50,000 per visit with an injection.

The majority of patients in Cohorts 1-1 and 2 reported that their relationship with their doctor was “good” (61% and 62%, respectively), whereas patients in Cohort 1-2 more frequently reported their relationship as “neither good nor bad” (59%). Patients in Cohort 1-2 were more likely to report a worse current understanding of AMD (23% as “not good”) than Cohorts 1-1 and 2 (6% and 2%, respectively; *p* = 0.0052), but the patients’ understanding at the start of treatment was comparable (Table 3). In Cohort 1, the patient–doctor relationship was correlated with the patient’s perceived treatment efficacy (*p* = 0.0013) and disease understanding (*p* = 0.0428) (Figure 2).

### 3.3. Visual Acuity

Table 4 shows the vision distribution between patients who agreed with the item “treatment was ineffective” and patients who disagreed with this item. Cohort 1 agreed with this item as a reason for discontinuation and Cohort 2 as a reason for dissatisfaction. The patient’s visual acuity was not correlated with the perceived treatment effect in Cohorts 1 (*p* = 0.6786) or 2 (*p* = 0.2644).

### 3.4. Factors Associated with Discontinuation or Dissatisfaction with Treatment

The reasons for discontinuing treatment were assessed in Cohorts 1-1 and 1-2 (Appendix A). The doctor’s decision was the top reason, because 89% of patients in Cohort 1 were advised by their doctor to stop (Figure 3a). Although the precise reason why the treatment was discontinued by the doctor was unavailable, “no deterioration in vision” (35.1%) was reported as the top reason in Cohort 1-1, and half of the patients in Cohort 1-1 were still visiting their doctor. By contrast, patients in Cohort 1-2 reported various reasons for discontinuation, with “no deterioration in vision” being the top reason. Almost half (45%) of patients in Cohort 1-2 felt unsure why they needed to continue treatment as compared with 9% in Cohort 1-1.

The top reason for dissatisfaction with anti-VEGF therapy in Cohort 2 was “time-consuming treatment with long waiting times” (77%) (Figure 3b), with agreement rates of >50% for “financial burden,” “ineffective treatment,” “feeling of the injection is unpleasant” and “scared of having an injection in the eye” (Appendix A).

### 3.5. Quality of Life

The overall results of the EQ-5D-5L imply that all cohorts maintained a healthy life (Table 5). Notably, the patients were more likely to report problems with pain/discomfort (31%) than the other dimensions of quality of life. Over half of the patients (55%) in Cohort 1-2 reported problems with pain/discomfort compared with Cohorts 1-1 and 2 (29% and 28%, respectively; *p* = 0.0422). The mean EQ-VAS score was slightly lower in Cohort 1-2 than Cohorts 1-1 and 2, but it was not significantly different between the three cohorts (*p* = 0.1340). However, the mean EQ-VAS scores were higher in patients who disagreed with the statement “treatment was not effective” than in patients who agreed with this statement (Cohort 1: 77.30 vs. 70.59, *p* = 0.0370; Cohort 2: 80.78 vs. 69.18, *p* = 0.0082).

### 3.6. Patient Activation

The mean ± standard deviation patient activation scores were numerically greater in Cohort 1-1 (51.8 ± 10.52, range 30.36–90.69) and Cohort 2 (51.5 ± 11.06, range 32.96–84.76) than in Cohort 1-2 (46.7 ± 10.31, range 30.36–70.15). About 5% of patients in Cohorts 1-1 and 2 were considered advocates for their own health (level 4) versus none in Cohort 1-2 (Table 6). Moreover, 55% of patients in Cohort 1-2 felt that their doctor is in charge of their health (level 1) as compared with 40% of patients in Cohort 1-1 and 48% in Cohort 2, but the gaps among cohorts were not significant. The mean activation score was higher in patients with a better understanding of nAMD than in patients with poor understanding, with mean values of 56.5, 48.9, 48.4 and 39.5 in patients reporting their understanding as “good”, “quite good”, “not quite good”, and “not good at all”, respectively (*p* < 0.001).

## 4. Discussion

This survey of patients with nAMD investigated the common reasons for discontinuing anti-VEGF therapy and the reasons for dissatisfaction with therapy. Most patients who discontinued treatment did so at their doctor’s decision (89%), and half of these patients were in a period of monitoring. These findings suggest that the doctor’s decision is a dominant factor in nAMD treatment discontinuation and that few patients with nAMD in Japan stop treatment at their own decision. Furthermore, half of the patients who stopped treatment at their doctor’s decision were still visiting their doctor, which is important considering that the management of nAMD is a long-term endeavor.

Although few patients discontinued at their own decision (11%; Cohort 1-2), the information obtained from this cohort, including the broad reasons for discontinuation, are clinically valuable. Patients in this cohort reported a poor understanding of nAMD and a worse relationship with the doctor compared with the other cohorts. They also reported more pain/discomfort than other cohorts. In real-world settings, patients who are more likely to drop out may have similar characteristics to Cohort 1-2 [13]. Such patients might require extra, ongoing attention and treatment from their doctor to help prevent a sudden deterioration in vision.

Patients in Cohort 1-1 discontinued treatment at the doctor’s decision and rarely acknowledged other reasons for discontinuation. However, half of the patients in Cohort 1-1 thought that their treatment was completed. We suggest that doctors should be more aware that, when they suggest discontinuation of treatment due to the absence of disease activity, they need to provide clear guidance to the patient about future follow-up plans.

Previous studies also showed that subjective dissatisfaction with the benefits of intravitreal injections is associated with treatment discontinuation [13,14,15,16]. In this survey, we found that the patient’s visual acuity was not correlated with the perceived treatment effect, which suggests that making the patient “feel” that their treatment is effective encompasses more than just the patient’s final vision. While the change in vision from baseline is a key factor, we also think that the following aspects are important for patient-perceived effectiveness: overall treatment experience combined with the patient’s satisfaction, the patient’s understanding of nAMD/treatment and the patient–doctor relationship. In fact, the patient–doctor relationship was correlated with the perceived treatment efficacy or disease understanding in patients who discontinued treatment. This raises the possibility that doctor–patient relationship is at least partly linked to the treatment outcomes, because patients with better treatment outcomes felt that they had a better relationship with their doctor, or vice versa. Providing sufficient explanation and support is vital for nAMD patients to ensure they have the right information about their eye health.

The burden of transport to hospital and periodic follow-up visits may be an important reason for discontinuation [14]. The burden of frequent visits and injections was noted by patients in Cohort 2. In Cohort 2, 42% and 48% of patients agreed with the reasons for frequent visits and injections, respectively; these perceptions could be negatively affected by long waiting times at each visit. Time-consuming treatment with long waiting times was the most frequently reported reason for dissatisfaction in this study. Other frequently reported reasons included “feeling of the injection is unpleasant” and “scared of having an injection in the eye.” When we designed the questionnaire, we took the former reason to signify the patient’s physical dislike of injections and the latter reason to signify the emotional stress caused by injections. Both reasons were selected by >50% of patients in Cohort 2. When Cohort 2 was divided into three groups by their treatment duration (≤1, 1 to ≤3 and ˃3 years), we found no marked differences in the perceptions of injections. Indeed, patients who continued treatment for many years still found injections to be scary and unpleasant, because 42% and 45% of patients who continued treatment for ˃3 years agreed with the reasons “scared of having an injection in the eye” and “feeling of the injection is unpleasant”, respectively. Patients in Cohorts 1 and 2 expressed concern regarding discomfort and fear. We assume that all patients were given topical anesthetic eyedrops prior to each injection, although this was not assessed in the questionnaire. Furthermore, topical anesthesia may not fully eliminate the sensation during injections, resulting in some discomfort, and may have contributed to the decisions to stop treatment or express dissatisfaction with treatment.

The present results also indicate that the financial burden of anti-VEGF therapy may be a barrier to ongoing treatment and was cited as a major reason for discontinuation and dissatisfaction. Patients reported out-of-pocket expenses of JPY ≥ 50,000 (approx. USD 450) per visit with an injection, which may be a considerable amount to pay repeatedly. In Japan, citizens/residents pay 10% to 30% of medical fees as out-of-pocket expenses depending on their age and income. Although the remainder is covered by public or private health insurance schemes, these out-of-pocket expenses may represent a significant barrier to ongoing treatment.

Although the present study was performed in the context of an online survey of Japanese patients, the general characteristics of these patients are broadly consistent with those of other studies in terms of their age, gender distribution and prevalence of comorbidities [17], supporting the generalizability of the findings. Our results are also consistent with those of previous studies (e.g., [15,18,19,20,21,22]). Our study revealed that the doctor’s decision was the predominant reason for stopping anti-VEGF therapy and that patients who discontinued at their own decision and those continuing treatment reported various reasons for discontinuation or dissatisfaction. Previous studies also indicated that treatment-related anxiety, financial considerations and transport burden placed on relatives or caregivers are important issues and that the ongoing, repeated treatment imposes a significant burden on patients [18]. Additionally, time-consuming treatment was noted as a significant burden in another study [21]. Therefore, new approaches or treatments that can reduce treatment burden are required.

Prior studies have documented the burden of AMD on quality of life [22,23]. In our study, the EQ-5D-5L data demonstrated relatively high quality of life in all cohorts, although the mean EQ-5D-5L index value and EQ-VAS were slightly lower in Cohort 1-2. One possible explanation for this is that the majority of patients in Cohort 1-2 reported hypertension as a comorbidity [23]. Most patients in Cohort 1-2 also had low activation levels, consistent with another study in which participants with lower PAM-13 scores missed more clinic visits [24]. The maximum PAM-13 score in Cohort 1-2 was lower than those in Cohorts 1-1 and 2 (70.15 vs. 90.69 and 84.76, respectively), and none of the patients in Cohort 1-2 were considered advocates of their own health. Although Cohorts 1-1 and 2 had higher PAM-13 scores than Cohort 1-2, the overall mean PAM-13 score (51.3) in this study was quite low relative to that in other studies [25,26].

The results must also be interpreted in consideration of the questions related to treatment discontinuation and possibility of patients refusing treatment. The treat and extend approach is widely used in Japan [27], as in other countries, for treating nAMD [28,29]. However, it is possible that some patients were on other regimens, including monthly/bimonthly and pro re nata. In this study, we defined discontinued using the question “Has it been 6 months or longer since you last received anti-VEGF therapy?”. Because we did not include a question regarding the administration regimen, it is possible that the treatment had been paused rather than discontinued in some patients, which may have been misunderstood in some patients and could be construed as a limitation. Additionally, we must also consider the possibility that some patients were continuing their follow-up visits but rejected administration of an anti-VEGF agent, perhaps due to discomfort from the injection or cost. Furthermore, we did not ask which anti-VEGF drug was being used. It has been reported that aflibercept is more widely prescribed than ranibizumab in Japan [30], but this could not be examined in the present study.

The use of the Rakuten Insight AMD panel and the online survey may also introduce some bias towards patients who are more web-literate and willing to participate in market research studies. Thus, the survey may be less representative of older patients, in particular, who may be less likely to participate in online surveys.

Other limitations include the possibility of selection bias because younger and relatively healthier patients may be more likely to be registered in an online patient panel. In fact, the health status of the respondents was good, as indicated by the EQ-5D-5L results. The age distribution of our cohorts was slightly younger than that of clinical trials, but was still reasonable. Another limitation is the possibility of recall bias, especially for events that happened months/years ago. It is possible that some respondents had diseases other than AMD and did not require/receive treatment with anti-VEGF agents. However, the general characteristics of the patients, including referral letters, out-of-pocket expenses and absence of diabetes, suggest that the criteria we used to select patients with AMD are appropriate.

In conclusion, we found that the top reason for discontinuing anti-VEGF therapy was the doctor’s decision. Patients who discontinued at the doctor’s decision rarely agreed with other reasons, whereas other patients who discontinued at their own volition agreed with multiple reasons. In addition, patients continuing anti-VEGF therapy reported various reasons for dissatisfaction, with “time-consuming treatment with long waiting times” being the top reason. As in previous studies, the burden of nAMD treatment was still high. Addressing these issues would help improve the clinical outcomes of anti-VEGF therapy in real-world settings. Providing sufficient explanation and patient-centered care seems vital for patients with nAMD to avoid unguided discontinuation and to ensure patients continue regular monitoring, even after pausing treatment.

## Figures and Tables

**Figure 1 jcm-10-03106-f001:**
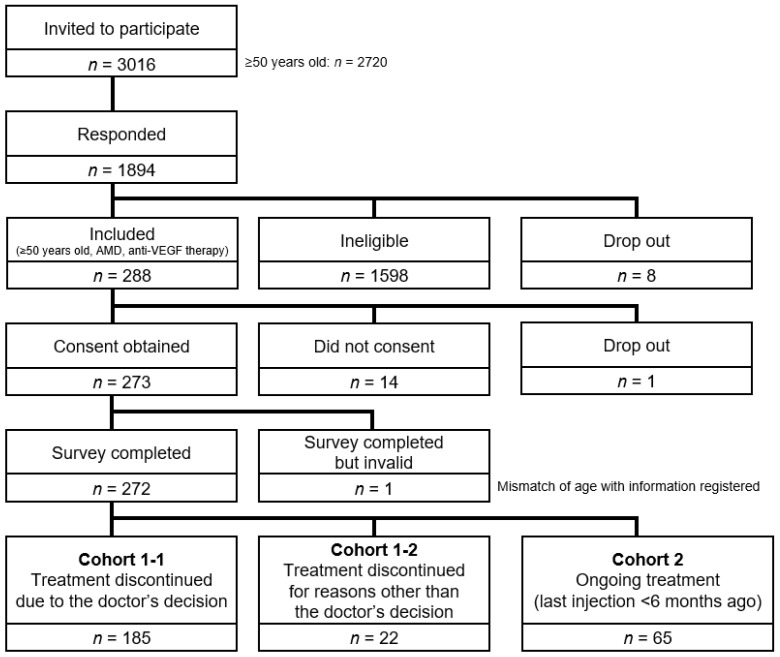
Patient disposition. The overall response rate was 62.8% (1894/3016) of those invited (69.6% of those aged ≥50 years; 1894/2720). Overall, 15.2% of patients who responded to the survey were included in the analysis (288/1894). Abbreviations: AMD, age-related macular degeneration; VEGF, vascular endothelial growth factor.

**Figure 2 jcm-10-03106-f002:**
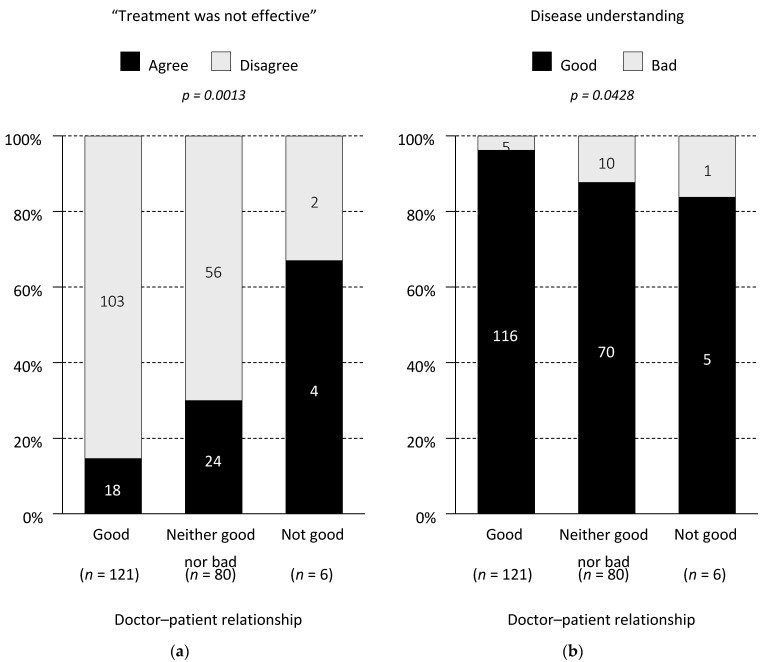
Doctor–patient relationship in Cohort 1. (**a**) Correlation between the doctor–patient relationship (good/neither good nor bad/not good) and perceived treatment efficacy. (**b**) Correlation between the doctor–patient relationship (good/neither good nor bad/not good) and disease understanding. The numbers on the bars indicate the number of respondents. The responses “quite good” and “not quite good” were combined as “good” or “bad”.

**Figure 3 jcm-10-03106-f003:**
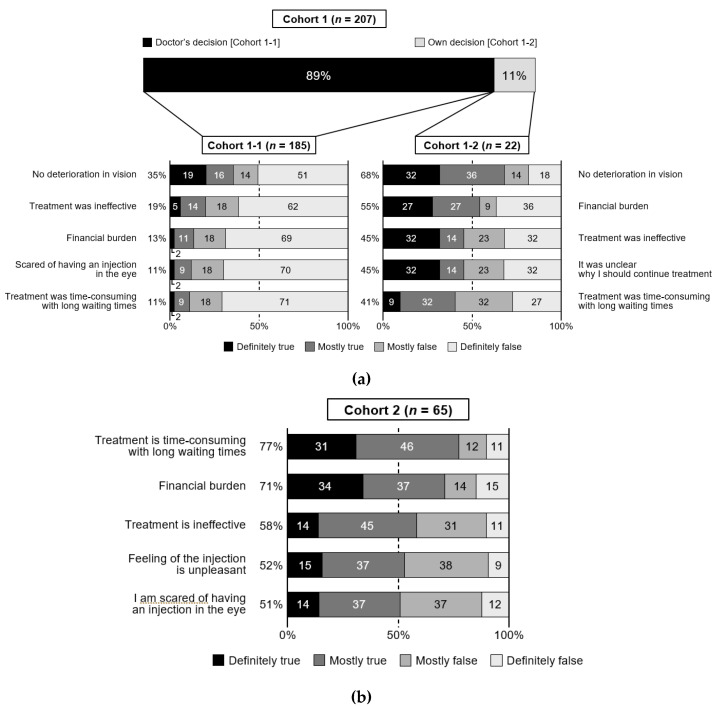
Top five reasons for treatment discontinuation in Cohort 1 (**a**) and dissatisfaction with treatment in Cohort 2 (**b**). Reasons are ordered in descending order according to the percentage of patients reporting “definitely true” or “mostly true”. The other reasons for discontinuation in Cohort 1 or dissatisfaction in Cohort 2 are listed in Appendix A, respectively. Cohort 1-1 patients discontinued at their doctor’s decision; Cohort 1-2 patients discontinued at their own decision; Cohort 2 patients were continuing treatment.

**Table 1 jcm-10-03106-t001:** Patient characteristics.

	Cohort 1-1 (Discontinued Due to the Doctor’s Decision)	Cohort 1-2 (Discontinued Due to Their Own Decision)	Cohort 2 (Continuing Treatment)
*n*	185	22	65
Sex			
Male	159 (85.9)	20 (90.9)	54 (83.1)
Female	26 (14.1)	2 (9.1)	11 (16.9)
Age, years			
50–59	48 (25.9)	6 (27.3)	9 (13.8)
60–69	72 (38.9)	9 (40.9)	22 (33.8)
70–79	60 (32.4)	6 (27.3)	30 (46.2)
≥80	5 (2.7)	1 (4.5)	4 (6.2)
Mean age	65.7	65.9	68.2
Living with family			
Yes	157 (84.9)	16 (72.7)	59 (90.8)
No	28 (15.1)	6 (27.3)	6 (9.2)
Family member			
Spouse/partner	140 (89.2)	15 (93.8)	57 (96.6)
Children	71 (45.2)	9 (56.3)	19 (32.2)
Other	28 (17.8)	3 (18.8)	5 (8.5)
Employment			
Yes	101 (54.6)	14 (63.6)	29 (44.6)
No	84 (45.4)	8 (36.4)	36 (55.4)
Smoking			
Everyday	28 (15.1)	6 (27.3)	8 (12.3)
Sometimes	1 (0.5)	1 (4.5)	1 (1.5)
Used to	108 (58.4)	11 (50.0)	36 (55.4)
Never	48 (25.9)	4 (18.2)	20 (30.8)
Comorbidity			
Glaucoma	15 (8.1)	0	2 (3.1)
Cataract	24 (13.0)	0	9 (13.8)
Stroke	5 (2.7)	1 (4.5)	2 (3.1)
Heart disease	16 (8.6)	3 (13.6)	6 (9.2)
Diabetes	32 (17.3)	6 (27.3)	13 (20.0)
Hypertension	71 (38.4)	13 (59.1)	22 (33.8)
Hyperlipidemia	22 (11.9)	4 (18.2)	8 (12.3)
Cancer	12 (6.5)	1 (4.5)	4 (6.2)
Other	33 (17.8)	2 (9.1)	12 (18.5)
None	44 (23.8)	5 (22.7)	22 (33.8)

Values are *n* (%).

**Table 2 jcm-10-03106-t002:** Disease onset and visual problems.

	Cohort 1-1 (Discontinued Due to the Doctor’s Decision)	Cohort 1-2 (Discontinued Due to Their Own Decision)	Cohort 2 (Continuing Treatment)
*n*	185	22	65
Diagnosis			
Within 1 year	2 (1.1)	0	6 (9.2)
Within 3 years	55 (29.7)	4 (18.2)	23 (35.4)
Within 5 years	59 (31.9)	7 (31.8)	16 (24.6)
Within 10 years	47 (25.4)	5 (22.7)	13 (20.0)
More than 10 years	22 (11.9)	6 (27.3)	7 (10.8)
Number of affected eyes			
One	166 (89.7)	21 (95.5)	56 (86.2)
Both	19 (10.3)	1 (4.5)	9 (13.8)
Current BCVA of the affected eye (self-reported)			
0.01–0.1 (20/2000 to 20/200)	44 (23.8)	4 (18.2)	9 (13.8)
0.2–0.3 (20/100 to 20/67)	26 (14.1)	4 (18.2)	11 (16.9)
0.4–0.6 (20/50 to 20/33)	39 (21.1)	5 (22.7)	22 (33.8)
0.7–1.0 (20/29 to 20/20)	53 (28.6)	6 (27.3)	11 (16.9)
≥1.2 (≥20/17)	12 (6.5)	2 (9.1)	8 (12.3)
Not sure	11 (5.9)	1 (4.5)	4 (6.2)
Difficulties in daily life			
Recognizing faces or objects	21 (11.4)	4 (18.2)	8 (12.3)
Reading books or newspapers	72 (38.9)	13 (59.1)	20 (30.8)
Driving	33 (17.8)	6 (27.3)	15 (23.1)
Going out alone	2 (1.1)	0	3 (4.6)
Other	21 (11.4)	2 (9.1)	7 (10.8)
No particular difficulties	81 (43.8)	3 (13.6)	30 (46.2)

Values are *n* (%); BCVA, best-corrected visual acuity.

**Table 3 jcm-10-03106-t003:** History of treatment and knowledge of AMD.

	Cohort 1-1 (Discontinued Due to the Doctor’s Decision)	Cohort 1-2 (Discontinued Due to Their Own Decision)	Cohort 2 (Continuing Treatment)
*n*	185	22	65
Type of hospital			
Eye clinic	42 (22.7)	10 (45.5)	13 (20.0)
Eye hospital with referral	123 (66.5)	8 (36.4)	47 (72.3)
Eye hospital without referral	20 (10.8)	4 (18.2)	5 (7.7)
Time to hospital			
<30 min	77 (41.6)	7 (31.8)	31 (47.7)
30–59 min	76 (41.1)	10 (45.5)	19 (29.2)
1 to <1.5 h	21 (11.4)	2 (9.1)	11 (16.9)
≥1.5 h	11 (5.9)	3 (13.6)	4 (6.2)
Method of transport to hospital			
Car	71 (38.4)	11 (50.0)	21 (32.3)
Taxi	2 (1.1)	1 (4.5)	1 (1.5)
Public transport	85 (45.9)	7 (31.8)	34 (52.3)
Bicycle	13 (7.0)	1 (4.5)	2 (3.1)
Foot	14 (7.6)	2 (9.1)	7 (10.8)
Accompanied by someone			
Yes	41 (22.2)	8 (36.4)	14 (21.5)
No	144 (77.8)	14 (63.6)	51 (78.5)
Treatment duration			
<3 months	55 (29.7)	7 (31.8)	4 (6.2)
≥3 months	52 (28.1)	7 (31.8)	7 (10.8)
≥1 year	37 (20.0)	5 (22.7)	23 (35.4)
≥3 years	24 (13.0)	3 (13.6)	16 (24.6)
≥5 years	17 (9.2)	0	15 (23.1)
Visit frequency			
Monthly	102 (55.1)	11 (50.0)	29 (44.6)
Every 2 months	37 (20.0)	5 (22.7)	24 (36.9)
Every 3 months	28 (15.1)	1 (4.5)	11 (16.9)
Every 6 months	7 (3.8)	2 (9.1)	0
None of the above	11 (5.9)	3 (13.6)	1 (1.5)
Injection frequency			
Monthly	57 (30.8)	4 (18.2)	10 (15.4)
Every 2 months	38 (20.5)	6 (27.3)	19 (29.2)
Every 3 months	27 (14.6)	1 (4.5)	25 (38.5)
Every 6 months	11 (5.9)	3 (13.6)	7 (10.8)
None of the above	52 (28.1)	8 (36.4)	4 (6.2)
Out-of-pocket payment (JPY)			
<10,000	38 (20.5)	5 (22.7)	9 (13.8)
10,000–29,999	40 (21.6)	7 (31.8)	26 (40.0)
30,000–49,999	50 (27.0)	5 (22.7)	22 (33.8)
≥50,000	57 (30.8)	5 (22.7)	8 (12.3)
Relationship with their doctor			
Very good	45 (24.3)	1 (4.5)	17 (26.2)
Good	68 (36.8)	7 (31.8)	23 (35.4)
Neither good nor bad	67 (36.2)	13 (59.1)	22 (33.8)
Not good	2 (1.1)	1 (4.5)	3 (4.6)
Not good at all	3 (1.6)	0	0
Understanding of AMD at the start of treatment			
Good	34 (18.4)	5 (22.7)	16 (24.6)
Quite good	78 (42.2)	8 (36.4)	22 (33.8)
Not quite good	49 (26.5)	6 (27.3)	17 (26.2)
Not good at all	24 (13.0)	3 (13.6)	10 (15.4)
Current understanding of AMD			
Good	58 (31.4)	4 (18.2)	29 (44.6)
Quite good	116 (62.7)	13 (59.1)	35 (53.8)
Not quite good	10 (5.4)	4 (18.2)	1 (1.5)
Not good at all	1 (0.5)	1 (4.5)	0

Values are *n* (%); AMD, age-related macular degeneration.

**Table 4 jcm-10-03106-t004:** Self-reported visual acuity of the affected eye and agreement with the response “treatment was ineffective”.

Self-Reported Visual Acuity	Cohort 1 (*n* = 207) (Discontinued Treatment)		Cohort 2 (*n* = 65) (Continuing Treatment)	
	Agree	Disagree	Agree	Disagree
*n*	46 (22.2)	161 (77.8)	38 (58.5)	27 (41.5)
0.01–0.3 (20/2000 to 20/67)	20 (25.6)	58 (74.4)	13 (65.0)	7 (35.0)
0.4–0.6 (20/50 to 20/33)	10 (22.7)	34 (77.3)	12 (54.5)	10 (45.5)
≥0.7 (≥20/29)	13 (17.8)	60 (82.2)	9 (47.4)	10 (52.6)
Not sure	3 (25.0)	9 (75.0)	4 (100.0)	0

Patients were asked about whether they agreed or disagreed with the response “Because the treatment was ineffective”. Values are *n* (%).

**Table 5 jcm-10-03106-t005:** Distribution of EQ-5D-5L.

	Cohort 1-1 (Discontinued Due to the Doctor’s Decision)	Cohort 1-2 (Discontinued Due to Their Own Decision)	Cohort 2 (Continuing Treatment)
*n*	185	22	65
EQ-5D-5L index value			
Mean (SD)	0.923 (0.107)	0.871 (0.173)	0.924 (0.111)
Median (range)	1.000 (0.505–1.000)	0.895 (0.285–1.000)	1.000 (0.605–1.000)
EQ-VAS score (0–100)			
Mean (SD)	76.5 (15.1)	69.7 (17.9)	74.0 (19.2)
Median (range)	80 (10–100)	75 (30–90)	80 (12–100)

SD, standard deviation.

**Table 6 jcm-10-03106-t006:** Distribution of patient activation levels.

	Cohort 1-1 (Discontinued Due to the Doctor’s Decision)	Cohort 1-2 (Discontinued Due to Their Own Decision)	Cohort 2 (Continuing Treatment)
*n*	176	20	65
PAM-13 score			
Mean (SD)	51.8 (10.52)	46.7 (10.31)	51.5 (11.06)
Median (range)	51.0 (30.36–90.69)	46.1 (30.36–70.15)	48.9 (32.96–84.76)
Activation level, *n* (%)			
Level 1 (0.0–47.0)	71 (40.3)	11 (55.0)	31 (47.7)
Level 2 (47.1–55.1)	40 (22.7)	4 (20.0)	12 (18.5)
Level 3 (55.2–72.4)	57 (32.4)	5 (25.0)	19 (29.2)
Level 4 (72.5–100.0)	8 (4.5)	0	3 (4.6)

SD, standard deviation.

## Data Availability

Data are contained within the article or Appendix A. Due to the proprietary nature of the patient panel and online survey, data from this study are not available to external researchers, beyond that reported in this manuscript.

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
