# Peer review of "Factors of Anti-Vascular Endothelial Growth Factor Therapy Withdrawal in Patients with Neovascular Age-Related Macular Degeneration: Implications for Improving Patient Adherence"

_jcm, 2021, doi:10.3390/jcm10143106_

Round 1
Reviewer 1 Report
In this study, the authors utilize an online survey to investigate reasons for discontinuation of anti-VEGF therapy in patients with neovascular AMD and associations with quality of life and patient activation. They divided respondents into three cohorts, those discontinuing treatment at the doctor’s discretion, those discontinuing treatment of their own volition, and those continuing treatment. The majority of patients discontinuing treatment was because of the doctor’s decision. Those discontinuing of their own volition tended to have worse relationships with their doctor, less understanding of the disease, and cited long wait times and cost.
This is a well thought out and executed survey study and has several clinically useful implications to ensure adherence to treatment.
It is somewhat surprising that a substantially greater percentage of patients were “discontinued” versus continuing to receive treatment. Perhaps it might be important to make the distinction between discontinuing treatment versus pausing treatment. The wording in the survey may have been confusing to patients. The treat and extend strategy is most commonly employed in the United States. It might be good to add into the discussion what is most commonly used in Japan and whether PRN strategy is predominant.
The low percentage of patients discontinuing anti-VEGF therapy at their own volition is a limitation of the study but perhaps also a reflection of higher patient adherence in Japan, which may limit generalizability of the study. In addition, a brief discussion of the healthcare system in Japan would be helpful to put cost of visits into context.
Here are a few additional points to consider:
- Is there information about the method of anesthesia used for the injections? This may impact the discomfort felt and may influence a patient’s decision to stop treatment.
- Is there information about the particular anti-VEGF drug being used? It would be interesting to see if those in Cohort 1-2 were being treated more often with a certain drug.
- Is the number of previous injections for each patient known?
Reviewer 2 Report
The paper can be improved as commented below:
- Some description of Rakuten AMD panel should be provided, so that readers can assess the generalizability of the survey findings. For example, if the Rakuten AMD panel are mainly consisted of well-educated patients or good socio-economic status patients, the survey results may not be generalized to the general AMD patients.
- At Figure 1, it is not clear among 3016 patients who were invited to participate, how many patients received anti-VEGF treatment for wet AMD thus were eligible for the study? Could you provided such information, and use this as a denominator for calculating the survey response rate?
- Cohort 2 definition should be made clear. Are cohort 2 consisted of all patients with ongoing anti-VEGF treatment or only the patients who had ongoing anti-VEGF treatment but were dissatisfied with treatment?
- In Table 1 or Table 2, could you provide information on the type of anti-VEGF patients received. Since some patients might change the type of anti-VEGF, the type of anti-VEGF of most recent injection can be used for reporting.
- Table 3, for the treatment duration, please make the categories mutually exclusive. E.g. “>=3 months” should be changed to “>=3 months to <1 year”
- In the table, please indicate visual acuity is based on patient self-report at the time of survey. For visual acuity, please provide the Snellen equivalent notation (20/x) as most of readers are more familiar with the Snellen Equivalent notation.
- Based on Table 1, look like about half of patients had good VA (20/40 or better). So the doctor’s decision for discontinue treatment might be appropriate. The patients who had worse VA (e.g., VA worse than 20/40) and discontinued treatment are more concerning and need further investigation of the reasons of treatment discontinuation.
- Table 6, the column for all patients combined probably can be deleted to be consistent with other tables.
Round 2
Reviewer 2 Report
Thanks for addressing all the previous comments.